# Can We Count on LLMs? The Fixed-Effect Fallacy and Claims of GPT-4 Capabilities

**Thomas Ball, Shuo Chen and Cormac Herley**
*Microsoft Research*
*Redmond, WA*

**Reviewed on OpenReview:** *https://openreview.net/forum?id=qt4dOEGZsK*

## Abstract

In this paper we explore evaluation of LLM capabilities. We present measurements of GPT-4 performance on several deterministic tasks; each task involves a basic calculation and takes as input parameter some element drawn from a large well-defined population (e.g., count elements in a list, multiply two k-digit numbers, etc). We examine several conditions per-task and perform enough trials so that statistically significant differences can be detected. This allows us to investigate the sensitivity of task-accuracy both to query phrasing and input parameter population. We find that seemingly trivial modifications in the task-prompt or input population can yield differences far larger than can be explained by sampling effects. For example, performance on a simple list-counting task varies with query-phrasing and list-length, but also with list composition (i.e., the thing-to-be-counted) and object frequency (e.g., success when an element accounts for $\approx 50\%$ of a list is different from when it accounts for $\approx 70\%$ etc).

We conclude that efforts to quantify LLM capabilities easily succumb to the language-as-fixed-effect fallacy, where experimental observations are improperly generalized beyond what the data supports. A consequence appears to be that intuitions that have been formed based on interactions with humans form a very unreliable guide as to which input modifications should "make no difference" to LLM performance.

## 1 Introduction

Rapid improvements in the performance of large language models (LLMs) have spurred great interest in evaluating their capabilities. In addition to answering general knowledge questions and summarizing text, GPT-4 has demonstrated the capability to compose poetry, solve chess puzzles and Geometry problems, and perform basic coding tasks. Capabilities that seem beyond the simple next-token-prediction they were trained on, causes some to suggest this as evidence of emergent behaviors from LLMs, or even that we may be witnessing the early signs of Artificial General Intelligence (AGI) (Bubeck et al., 2023). Others are not convinced, and suggest that LLMs simply parrot pastiches of text snippets from their training sets (Bender et al., 2021).

The documentation of surprising capabilities has been accompanied by many accounts of failures. Hallucinations (where LLMs offer plausible but entirely invented detail) have proved hard to eliminate. Arkoudas points out that GPT-4 struggles with some basic tasks that humans find easy or trivial; e.g., they aren't reliable even on tasks such as counting, multiplication, etc (Arkoudas, 2023). McCoy et al suggest that many of the remarkable capabilities are simply artifacts of the training set and autoregressive task that GPT-4 was trained to solve (McCoy et al., 2023).

An accumulation of observed successes and failures at particular tasks unfortunately does little to settle questions about LLM reliability or capabilities. In this paper we present results on a series of deterministic

tasks; each of the tasks involves a basic calculation and takes as input parameter some element drawn from a large well-defined population (e.g., count elements in a list, multiply two $k$-digit numbers, etc). Since, by construction, the correct answer is easy to determine, we can measure performance without costly and subjective hand-labelling or assessments. By randomly sampling the input parameter populations we can measure performance on large numbers of that are semantically and logically equivalent. Since the parameter spaces can be arbitrarily large the concern about verbatim contamination of training data is greatly reduced. This allows us to investigate the sensitivity of task-accuracy both to query phrasing and input parameter population; we do this at sufficient scale to detect statistically significant differences. We investigate both re-wordings of the prompt and changes to the input population. For example, our population might be length-21 lists of floating point numbers, and the task might be to find the median, but modifications might be to try reworded versions of the prompt, or try lists with a different number of significant decimal places given.

Our contributions are as follows. We present measurements of GPT-4 performance on several deterministic tasks. We examine several conditions per-task and perform enough trials (500 per condition unless otherwise stated) so that statistically significant differences can be detected. For all tasks and all conditions this entails about 37k responses from GPT-4; all prompts, responses and associated metadata are openly available to those who wish to check or build upon our findings. We measure performance on tasks such as counting, sorting, multiplication, etc, and find that accuracy, while better-than-random, is often very poor. We find that seemingly trivial modifications both in the prompt-phrasing and parameter population can yield differences far larger than can be explained by sampling effects. For example, performance on a simple list-counting task varies with query-phrasing and list-length, but also with list composition (i.e., the thing-to-be-counted) and object frequency (e.g., success when an element accounts for $\approx 50\%$ of a list is different from when it accounts for $\approx 70\%$ etc).

We conclude that efforts to quantify LLM capabilities easily succumb to the language-as-fixed-effect fallacy (Clark, 1973; Coleman, 1964; Yarkoni, 2022), where experimental observations on language-tasks are improperly generalized beyond what the data supports. A consequence appears to be that intuitions that have been formed based on interactions with humans form a very unreliable guide as to which modifications should "make no difference" to LLM performance. For example, the abstractions that we take for granted for humans (e.g., of separating the task of counting from the thing-to-be-counted) do not appear to be replicated by LLMs.

Sensitivity of LLM performance to query phrasing has spawned efforts to improve accuracy by using few-shot, Chain-of-Thought and scratchpad techniques. However, efforts to quantify this sensitivity are nascent. Sclar et al examine the effect of phrasing on accuracy for multiple choice tasks using the LLaMA-2-13B model (Sclar et al., 2023). Sun et al examine zero-shot robustness for the MMLU (Hendrycks et al., 2020) and BIG-bench (Ghazal et al., 2013) datasets using several models having between 3B and 13B parameters. There are important points of difference between ours and previous work. First, we explore accuracy on atomic tasks such as counting and multiplication rather than on datasets of multiple-choice questions that may have been seen in training (e.g., there is evidence that GPT4 has seen the BIG-bench canary GUID (Bubeck et al., 2023)). Second, we use parameterized tasks and explore sensitivity to input parameters as well as prompt phrasing (e.g., showing that counting accuracy depends on the thing-to-be-counted). Finally, we evaluate using GPT-4; this has between one and two orders of magnitude more parameters than those used in (Sclar et al., 2023; Sun et al., 2023). This allows us to have confidence that the problems do not seem significantly alleviated by model scale.

We wish to be clear that our goal is not to determine whether LLMs can or cannot count, sort, or multiply, etc. First, we have other ways of performing these tasks. Second, it is possible that prompt engineering, providing few-shot examples, the use of Chain-of-Thought reasoning, or the invocation of plug-ins might sometimes improve performance. However, our goal is not to improve the accuracy in particular settings. Rather, it is to draw attention to an unaddressed difficulty in establishing accuracy: evaluation of LLM capabilities seems particularly susceptible to a major pitfall that exists when we go from particular experimental observations to general claims. That is, sensitivity to seemingly trivial modifications means that observed accuracy numbers cannot be assumed to generalize (even to entirely equivalent versions of a task). So, for example, while prompt engineering might yield accuracy improvement on a particular version of a task, we can't assume that

that improvement will be observed in rephrased versions. While we've demonstrated the problem on basic arithmetic tasks it seems unlikely to be confined to that domain. For example, LLM performance at certain tasks might be improved by invoking a plug-in, writing code or using Chain-Of-Thought, but deciding when and how to do so is itself a task with success rate subject to the sensitivities we highlight. That is, invoking plugins doesn't solve even the basic counting task if the decisions on when and which plugin to invoke is itself brittle and sensitive to prompt phrasing. Finally, we acknowledge that GPT-4 is the largest and most recent model we tested; an interesting direction for future work would be to determine whether the problem is reduced in later models.

So, can LLMs count (or multiply, or sort etc)? Our evidence suggests that variation as we sample possible phrasings is too high to allow a Yes-or-No answer, and that accuracy estimates must be regarded as particular to the experimental setup used. This also means that reporting observed performance or accuracy numbers on other deterministic tasks (such as standardized tests (Katz et al., 2023; Takagi et al., 2023; Nori et al., 2023), textbook problems, etc) is not sufficient to establish general capabilities.

## 2 Background: The Language-as-Fixed-Effect Fallacy

The Language-as-Fixed-Effect Fallacy, as described by Clark (Clark, 1973), is the phenomenon where a claim supported by statistical evidence does not generalize beyond the specifics of the experimental setup. He illustrates with a language-task thought-experiment originally proposed by Coleman (Coleman, 1964). Let $N$ be the set of all English nouns, $V$ the set of all verbs, and let $T(.)$ be a test statistic representing how well humans perform at some task involving words (e.g., how well they can spell them, how quickly they can type them, etc). Suppose that experimenter A wishes to test the hypothesis that people perform the task better on nouns than on verbs:

$$\mathcal{H}_A = T(\boldsymbol{N}) > T(\boldsymbol{V}).$$

Suppose experimenter B wishes to test the opposite:

$$\mathcal{H}_B = T(\boldsymbol{N}) < T(\boldsymbol{V}).$$

Let's stipulate, by contrast, that they are both wrong, and that $T(\boldsymbol{N}) = T(\boldsymbol{V})$.

As a test of $\mathcal{H}_A$ the first experimenter selects subsets $\boldsymbol{N}_A \subset \boldsymbol{N}$ and $\boldsymbol{V}_A \subset \boldsymbol{V}$ each with some fixed number of randomly selected nouns and verbs. With this choice she recruits participants and on finding that $T(\boldsymbol{N}_A) > T(\boldsymbol{V}_A)$, by a statistically significant amount, she rejects the null hypothesis (that there's no difference) and concludes she has firm evidence in favor of $\mathcal{H}_A$. Similarly, the second experimenter selects at random different subsets $\boldsymbol{N}_B \subset \boldsymbol{N}$ and $\boldsymbol{V}_B \subset \boldsymbol{V}$ with fixed numbers of nouns and verbs. With this fixed choice he recruits participants and finds $T(\boldsymbol{N}_B) < T(\boldsymbol{V}_B)$, by a statistically significant amount, and concludes this is firm evidence for $\mathcal{H}_B$.

The problem is that while both A and B intend to generalize to the whole population $\boldsymbol{N}$ and $\boldsymbol{V}$ they have tested only on particular subsets. There is good evidence to believe that, with any collection of participants, we could verify both $T(\boldsymbol{N}_A) > T(\boldsymbol{V}_A)$ and $T(\boldsymbol{N}_B) < T(\boldsymbol{V}_B)$, but neither of these is enough to support either $\mathcal{H}_A$ or $\mathcal{H}_B$. In the language of statistical testing our experimenters have treated random effects as fixed (Clark, 1973).

Fixed effects are those that are considered constant across the relevant population, while random effects are those that vary (for an account of various other definitions see (Gelman, 2005)). In the experiments above there are two populations involved: the populations of noun-verb collections, and the population of human participants. When she generalized from $\boldsymbol{N}_A, \boldsymbol{V}_A$ to $\boldsymbol{N}, \boldsymbol{V}$ our first experimenter implicitly assumed that any other subsets $\boldsymbol{N}_C \subset \boldsymbol{N}$ and $\boldsymbol{V}_C \subset \boldsymbol{V}$ would also give the result that she observed (i.e., $T(\boldsymbol{N}_C) > T(\boldsymbol{V}_C)$). If this were true she'd be justified in thinking that her observed difference was powerful evidence for $\mathcal{H}_A$. If this is not true then her experiment supports only the narrow uninteresting claim $T(\boldsymbol{N}_A) > T(\boldsymbol{V}_A)$. Effectively, she assumed that what she observed wasn't particular to $\boldsymbol{N}_A, \boldsymbol{V}_A$ but general to $\boldsymbol{N}, \boldsymbol{V}$.

In a colloquial sense fixed effects are ones where the particular choice doesn't affect the generality we wish to claim. We expect, for example, that what an experimenter had for breakfast or what color socks she

was wearing has no effect on the outcome; these are not details that have to be faithfully reproduced to ensure replication of the original experiment. In this telling the fixed-effect fallacy is simply assuming that certain details don't matter when in fact they do. Unfortunately, there's no simple way to determine that a certain variable has no influence on an experimental result; experiments necessarily involve many judgements about which details matter and which do not, and many of those judgements are subjective. One of our findings is that intuitions about which modifications might make a difference can be very flawed; that human performance remains constant under a certain modification is no guarantee at all that LLM performance also will.

## 3 The Fixed-Effect Fallacy and LLM Task Performance

We wish to evaluate whether, and how well, an LLM can perform a particular task that has a single deterministic correct answer (e.g., counting, deciding to invoke a plug-in, or Retrieval-Augmented Generation etc). For the counting task one approach might be to produce a list of objects and prompt the LLM to count the occurrences of a particular item. To make the experimental setup concrete we might specify a list length and dictionary of possible elements. For example:

```
rLen = 20
listOfItems = ['mango','peach']
r = random.choices(listOfItems, k = rLen)
```

is a Python snippet that will return a length-20 list with the elements of `listOfItems` chosen at random with replacement. When there are only two elements, as shown, there's a population of $2^{20}$ such lists; call this population $\mathbf{R}$. We might prompt the LLM with:

```
prompt = ''How many times does 'mango' appear in this list: '' + str(r)
```

where $r \in \mathbf{R}$. By repeating this query with many different elements of $\mathbf{R}$ we might try to build a picture of the LLM's performance at the task.

In this setup choice of list from $\mathbf{R}$ is being treated as the only random effect; i.e., the only source of variation (Gelman & Hill, 2006). We are testing how well the LLM does over many different members of $\mathbf{R}$ but are assuming that other factors we might vary make no difference. However, there are many other populations of lists that we might try, and there are many other wordings of the prompt that could be used. If we use observed success with the above prompt to conclude that our LLM can count elements of a length-20 list with a particular success rate we are implicitly assuming that these other possible choices would make no difference. For example, an alternative to the prompt above might be:

```
prompt = ''Here is a list: '' + str(r) + ''. How many times does 'mango' appear on it?''
```

This would appear to be an equivalent evaluation of the task, or a modification that should make no difference. Unfortunately, this is not the case.

As we show in Section 4.2 these assumptions most definitely do not hold. Wording of the prompt and choice of the particular items to be counted can make a substantial difference to the answer (see Table 1). For example, the hypothesis that tests using the two prompts given above (with everything else held constant) produce results drawn from the same distribution, is robustly rejected by a $\chi^2$ test. Thus, if we report that our LLM can count with a particular success rate we are committing the same fixed-effect error as experimenters A and B above.

When we encounter a particular experimental result (e.g., $q = 0.86$ (86.0%) on the $N = 10$ counting task in Table 1) we generally understand that this involves some margin of error. For example, rather than $q$, we expect a repeat of the experiment to produce an estimate $q \pm \delta_q$. A very familiar case exploits the fact that 95% of the values of a normal distribution lie within 1.96 standard deviations of the mean, so we can write $\delta_q = 1.96 \cdot \sqrt{q \cdot (1 - q)/N}$ and be confident that 95% of trials will fall in this interval (Taylor & Thompson, 1982).

However, it is important to keep track of the baked-in assumptions: this estimate assumes that variance from sampling the list population (i.e., sampling $R$) is the only source of randomness. If significant other sources of randomness exist, then we know only that $\delta_q$ is greater than (and possibly much greater than) $1.96 \cdot \sqrt{q \cdot (1-q)/N}$. That is, we have only a lower bound on our margin-of-error. We can't rule out, for example, that the 95% confidence interval is $\pm 30\%$. The results of Section 4 show that other sources of randomness that are too large to ignore do exist for several of the tasks we consider (and in some cases appear far greater than the variance due to sampling).

## 4 Tasks

### 4.1 Experimental setup

In order to test LLM performance we choose tasks that have deterministic answers, and where it is relatively easy to decide if the LLM gives the correct answer. This obviates the need for subjective assessments, heuristics, hand-labelling or error-prone parsing of the response, and allows us to scale-up testing. The tasks we examine are: counting, finding the maximum, median and sorted version of a list of numbers, and long multiplication. The difficulty with counting and long multiplication has been observed by others (Arkoudas, 2023).

Unless otherwise specified all of the conditions were evaluated on 500 independent runs. Thus, for example, if a table entry reports a success rate of 89.0% on a task, and sampling were the only source of randomness, then a reasonable estimate of the 95% confidence interval would be $1.96 \cdot \sqrt{0.89 \times 0.11/500} \approx 2.74\%$. However, an important finding, below, is that there are significant other sources of randomness, and the conventional way of estimating margins-of-error cannot be applied. Lists were generated independently for each trial at query-time; thus, we did not re-use lists across conditions. All of the trials are performed using the OpenAI GPT-4 API with a temperature setting of 0.7. The results of all queries are available in the GitHub repository `https://github.com/demarinaGit/canWeCountOnLLMs`. All trials were performed using a temperature setting of 0.9.

For all of the tasks we give an example prompt together with the correct answer and GPT-4's answer. Due to space constraints we show only examples where the GPT-4 response is incorrect. This is not reflective of its accuracy: in each case we give a table showing how accuracy evolves with problem size. However, in giving examples where the answers are incorrect we illustrate that they are often very significantly better-than-random.

### 4.2 Count

First we examine the capability of GPT-4 to perform basic counting tasks. We choose a length-`rLen` list with two possible elements and ask GPT-4 to count the number of occurrences of the first element. An example query is (let's call this wording #1):

> **Prompt:** How many times does 'mango' appear in this list: [mango, peach, peach, peach, mango, mango, mango, peach, peach, peach, mango, mango, mango].
> **Correct Answer:** 7
> **GPT-4 Answer:** 'Mango' appears 6 times in this list.

We evaluate for five different target lengths; the results are shown in the first column of Table 1. In choosing modifications of this task we choose a different variations of the input list by replacing the word-pair 'mango/peach' with 'airedale/aspidistra' (results in column 2). We alter the weights: i.e., have 'mango' and 'peach' appear with probabilities 70% and 30% instead of 50% and 50% (results in columns 3). We also examine one simple rewording of the prompt (let's call this wording #2):

> **Prompt:** Here is a list: [mango, peach, peach, peach, mango, mango, mango, peach, peach, peach, mango, mango, mango]. How many times does 'mango' appear on it?

> **Correct Answer: 7**
> **GPT-4 Answer:** 'Mango' appears 6 times in this list.

This gives us a total of four conditions, all of which involve the same basic counting task. We evaluate each condition with list lengths `rLen`= 10, 15, 20, 30 and 40, and we perform 500 trials per condition. The results are shown in Table 1. Thus, the five rows and first four columns represent a total of $5 \times 4 \times 500 = 10,000$ queries to GPT-4.

| rLen | Wording #1 Wts=[0.5,0.5] mango/peach | Wording #1 Wts=[0.5,0.5] airedale/ aspidistra | Wording #1 Wts=[0.7,0.3] mango/peach | Wording #2 Wts=[0.5,0.5] mango/peach | Comp. Cols(1,2) $(\chi^2, p)$ | Comp. Cols(1,3) $(\chi^2, p)$ | Comp. Cols(1,4) $(\chi^2, p)$ |
|---|---|---|---|---|---|---|---|
| 10 | 89.0% | 91.2% | 70.2% | 96.6% | (1.12, 2.9e-01) | **(53.26, 2.92e-13)** | **(20.49, 6.e-06)** |
| 15 | 61.2% | 53.6% | 31.8% | 88.6% | **(5.6, 1.8e-02)** | **(85.68, 2.11e-20)** | **(98.38, 3.45e-23)** |
| 20 | 48.2% | 29.6% | 30.8% | 76.2% | **(35.61, 2.41e-09)** | **(30.95, 2.65e-08)** | **(82.18, 1.24e-19)** |
| 30 | 12.4% | 7.4% | 19.0% | 43.6% | **(6.46, 1.10e-02)** | **(7.74, 5.41e-03)** | **(119.17, 9.60e-28)** |
| 40 | 12.6% | 7.6% | 17.6% | 21.0% | **(6.34, 1.18e-02)** | **(4.49, 3.40e-02)** | **(12.03, 5.25e-04)** |

Table 1: Percent correct for counting the occurrences of a length-`rLen` list with two items chosen uniformly-at-random. Performance decays rapidly with list length. On the right-hand side of the table we present $\chi^2$ tests comparing the results of the first condition with each of the others. This test evaluates the null hypothesis that the answers in the various conditions are drawn from the same distribution. Boldface entries are cases where $p < 0.05$ and we reject the null hypothesis. The null hypothesis is robustly rejected for almost all lengths and conditions. E.g., when comparing columns 1 and 4 (i.e., simply switching between wording #1 and wording #2 with the 'mango/peach' word-pair). This demonstrates that simple modifications of the task (that might easily be assumed to make no difference) in fact are sources of variance beyond what can be explained by sampling effects.

We use a $\chi^2$ test to determine if the responses to different ways of phrasing the task are drawn from the same distribution. For example, we can take the null hypothesis to be that some row of the first and fourth columns of Table 1 represent answers drawn from the same distribution. E.g, for `rLen`= 10 there were $445/500$ and $483/500$ correct trials respectively. Using a standard $\chi^2$ test to compare these two distributions of correct/incorrect answers yields ($\chi^2 = 20.49$, $df = 1$, $p = 6.0e - 6$). The $p$-value can be taken as an estimate of the probability of these results being observed if columns 1 and 4 of row 1 were produced by the same process; generally when $p < 0.05$ we say that the null hypothesis is rejected. Similarly for all the other rows, the hypothesis (that results of the task with different wording are drawn from the same distribution) is rejected. The degrees-of-freedom is $df = 1$ for all of our tests since we are always doing pairwise comparisons on tasks on a binary outcome (Taylor & Thompson, 1982).

The results of our $\chi^2$ tests are given in the right-hand side of Table 1. The null hypothesis is robustly rejected for all lengths when comparing columns 1 and 4 (i.e., simply switching between wording #1 and wording #2 with the 'mango/peach' word-pair). The null hypothesis is rejected for several lengths when comparing columns 1 and columns 2, 3 (i.e., simply switching the word-pair while using wording #1). This demonstrates that simple modifications of the task (that might easily be assumed to make no difference) in fact are sources of variance beyond what can be explained by sampling effects.

We note also that the GPT answers are biased toward under-counting. For example in the 'mango/peach' case the mean of the correct answers for the five lengths tested (i.e., `rLen`= 10, 15, 20, 30 and 40) were: $(5.57, 7.96, 10.57, 15.46, 20.6)$ and the GPT-4 answers were $(5.45, 7.57, 10.04, 14.09, 18.5)$. Thus, across 500 trials, the mean GPT-4 answers were always lower. Among the 500 trials the ratio of over-counts:under-counts was $(55 : 0, 192 : 2, 248 : 11, 428 : 10, 451 : 11)$.

In the appendix we give results using GPT-3.5, Mistral7B, and Llama7B. Note that the same basic pattern holds: i.e., the different conditions

### 4.3 Maximum, Median and Sort

Here we ask GPT-4 to perform elementary tasks on lists of numbers: return the maximum, median and sorted version of the list. We evaluate three different conditions. First we ask for the maximum (or median or sorted version) of a list of `rLen` numbers drawn uniformly-at-random from the interval $(100.0, 20000.00)$ and rounded to two decimals places. An example of the prompt for the median-finding task is:

**Prompt:** What is the median value in this list: [7176.36, 5222.86, 1089.62, 19927.36, 5655.72, 18355.58, 18978.7, 7028.49, 14190.57, 14243.69, 11251.69]. Please write 'Answer='
**Correct Answer:** 11251.69
**GPT-4 Answer:** 7176.36

Second, we repeat with integers drawn uniformly-at-random from $(10, 99)$ (i.e., all list-members are 2-digit numbers). Finally, we use a list of `rLen` name-value pairs, where a randomly-chosen name is associated with a number drawn uniformly-at-random from the interval $(100.0, 20,000.00)$ and rounded to two decimals places. An example of the latter query is:

**Prompt:** Please sort this list in ascending order: [John: \$12158.21, Mary: \$1416.51, Peter: \$7507.58, Vivek: \$10941.54, Xian: \$10530.84, Alex: \$1641.14, Maria: \$1025.49, Frank: \$260.85, Luis: \$7464.35, Manuel: \$1782.86, Kristen: \$10085.24].
**Correct Answer:** [Frank: \$260.85, Maria: \$1025.49, Mary: \$1416.51, Alex: \$1641.14, Manuel: \$1782.86, Luis: \$7464.35,Peter: \$7507.58, Kristen: \$10085.24, Xian: \$10530.84, Vivek: \$10941.54, John: \$12158.21]
**GPT-4 Answer:** [Frank: \$260.85, Maria: \$1025.49, Mary: \$1416.51, Alex: \$1641.14, Manuel: \$1782.86, Peter: \$7507.58, Luis: \$7464.35, Kristen: \$10085.24, Xian: \$10530.84, Vivek: \$10941.54, John: \$12158.21]

The results of the maximum, median and sorting tasks are given in Tables 2, 3 and 4 respectively. The three different list conditions are explored in columns 1-3 of these tables. As in Section 4.2, we use a $\chi^2$ test to explore whether these different variations on the task produce answers that appear drawn from the same distribution. The right-hand portion of Tables 2, 3 and 4 gives the results; we do $\chi^2$ tests to compare columns 2 and 3 with column 1.

Table 2 shows the results of the maximum-finding task. Performance in all conditions is good, though not perfect (e.g, results are almost always $> 90.0\%$). The $\chi^2$ tests show that the hypothesis that performance on the name-value version of the list is consistent with performance on the value-only list is rejected for lengths $> 11$. The hypothesis that performance on the integer version of the list is consistent with performance on the 2-decimal floats list is rejected for all lengths.

Table 3 shows the results of the median-finding task. Performance in all conditions is poor (e.g, results are $< 90.0\%$). The $\chi^2$ tests show that the hypothesis that performance when the numbers are drawn from $(10.0, 20000.0)$ is consistent with performance when numbers are drawn as integers from $(10, 99)$ is rejected for all lengths. The hypothesis that that name-value version of the list is consistent with performance on the value-only list is also rejected for all lengths. Note that the $p$-values in both cases are $\ll 0.05$, so the probability that the same process accounts for both conditions is very low.

Table 4 shows the results of the sorting task. Performance in condition 2 is good, but is very poor in condition 3 (e.g, results in column 3 are $< 55.0\%$). The $\chi^2$ tests show that the hypothesis that performance when the numbers are drawn from $(10.0, 20000.0)$ is consistent with performance when numbers are drawn from $(10, 99)$ is rejected for all lengths. The hypothesis that that name-value version of the list is consistent with performance on the value-only list is also rejected for all lengths. Again, the $p$-values indicate robust rejection of these hypotheses.

| rLen | `float` in $(100.0, 20000.0)$ | `int` in $(10, 99)$ | Name-value `float` in $(100.0, 20000.0)$ | Compare Cols(1,2) $(\chi^2, p)$ | Compare Cols(1,3) $(\chi^2, p)$ |
|------|-------------------|------------|-----------------------|-----------------------|-----------------------|
| 11 | 97.79% | 100.0% | 97.2% | **(9.23, 2.38e-03)** | (0.16, 6.93e-01) |
| 15 | 96.4% | 100.0% | 92.2% | **(16.35, 5.27e-05)** | **(7.44, 6.37e-03)** |
| 21 | 94.4% | 100.0% | 86.6% | **(26.79, 2.27e-07)** | **(16.8, 4.16e-05)** |

Table 2: Comparison of the find-maximum task. The prompt simply asks GPT-4 to find the maximum of a list of numbers. Column 1: numbers uniform on $(100.0, 20000.0)$ to 2 decimals, Column 2: numbers uniform on $(10, 99)$ as integers, Column 3: name-value pairs with values uniform on $(100.0, 20000.0)$ to 2 decimals. The right-hand side of the table shows $\chi^2$ tests comparing Column 1 to each of the others. Boldface entries are cases where $p < 0.05$ and we reject the null hypothesis (that results in the given columns are produced by the same process). The null hypothesis is rejected except for length-11 when comparing columns #1 and #3: thus simply switching the list from numbers to name-value pairs introduces variance beyond what can be explained by sampling effects.

| rLen | `float` in $(100.0, 20000.0)$ | `int` in $(10, 99)$ | Name-value `float` in $(100.0, 20000.0)$ | Compare Cols(1,2) $(\chi^2, p)$ | Compare Cols(1,3) $(\chi^2, p)$ |
|------|-------------------|------------|-----------------------|-----------------------|-----------------------|
| 11 | 68.4% | 85.0% | 89.6% | **(37.62, 8.57e-10)** | **(66.46, 3.58e-16)** |
| 15 | 52.8% | 74.0% | 89.6% | **(47.51, 5.47e-12)** | **(163.32, 2.13e-37)** |
| 21 | 35.87% | 62.73% | 65.6% | **(65.82, 4.94e-16)** | **(87.12, 1.02e-20)** |

Table 3: Comparison of the find-median task. The prompt simply asks GPT-4 to find the median of a list of numbers. Column 1: numbers uniform on $(100.0, 20000.0)$ to 2 decimals, Column 2: numbers uniform on $(10, 99)$ as integers, Column 3: name-value pairs with values uniform on $(100.0, 20000.0)$ to 2 decimals. The right-hand side of the table shows $\chi^2$ tests comparing Column 1 to each of the others. Boldface entries are cases where $p < 0.05$ and we reject the null hypothesis (that results in the given columns are produced by the same process). The null hypothesis for all lengths and conditions: thus simply changing the range on the numbers, or switching to name-value pairs introduces variance beyond what can be explained by sampling effects.

| rLen | float in $(100.0, 20000.0)$ | int in $(10, 99)$ | Name-value float in $(100.0, 20000.0)$ | Compare Cols(1,2) $(\chi^2, p)$ | Compare Cols(1,3) $(\chi^2, p)$ |
|---|---|---|---|---|---|
| 11 | 94.93% | 99.77% | 52.0% | **(18.39, 1.80e-05)** | **(231.64, 2.62e-52)** |
| 15 | 94.75% | 100.0% | 36.0% | **(24.48, 7.50e-07)** | **(375.91, 9.69e-84)** |
| 21 | 88.32% | 99.8% | 15.0% | **(56.26, 6.34e-14)** | **(528.43, 6.2e-117)** |

Table 4: Comparison of the list-sorting task. The prompt simply asks GPT-4 to sort a list of numbers in ascending order. Column 1: numbers uniform on $(100.0, 20000.0)$ to 2 decimals, Column 2: numbers uniform on $(10, 99)$ as integers, Column 3: name-value pairs with values uniform on $(100.0, 20000.0)$ to 2 decimals. The right-hand side of the table shows goodness-of-fit $\chi^2$ tests comparing Column 1 to each of the others. Boldface entries are cases where $p < 0.05$ and we reject the null hypothesis (that results in the given columns are produced by the same process). The hypothesis that Column 2 or 3 is produced by the same process as Column 1 is rejected for all lengths: thus simply changing the range on the numbers, or switching to name-value pairs introduces variance beyond what can be explained by sampling effects.

### 4.4 Long Multiply

Here we evaluate performance at long multiplication, where we prompt the LLM to calculate the product of a $k_1$-digit by a $k_2$-digit number. An example for $4 \times 4$ is:

> **Prompt:** What is the product of 6438 and 9038? Please write 'Answer ='
> **Correct Answer:** 58186644
> **GPT-4 Answer:** Answer = 58169844.

Table 5 shows the performance multiplying a $k_1$-digit by a $k_2$-digit number for $k_1, k_2 \in \{2, 3, 4, 5\}$. Apart from the $2 \times 2$ case the results are largely poor. Observe that perfect performance on the $2 \times 2$ task drops to negligibly correct answers for $4 \times 4$.

Since there is sometimes a significant difference between the $k_1 \times k_2$ result with the $k_2 \times k_1$ result we perform a $\chi^2$ test on several of the off-diagonal elements. The results are shown in Table 6. Note that results for the $4 \times 2$ and $2 \times 4$ are significantly different, as are those for $5 \times 2$ and $2 \times 5$. Thus, even the hypothesis that performance on the $k_1 \times k_2$ multiplication will be equivalent to the $k_2 \times k_1$ is rejected for at least some lengths.

Both Dziri et al (Dziri et al., 2023) and Arkoudas (Arkoudas, 2023) look at the example of long multiplication. Dziri et al note that while the answers for $4 \times 4$ are almost always incorrect, the first and last two digits of the GPT-4 answers are almost always correct. They describe this as a matching of "surface probabilities." That is, the first two digits of a product are determined by the leading digits of the multiplicands irrespective of length. Thus, this portion of the answer can always be determined without paying attention to the rest. Similarly for the last few digits.

| $k_1$ \ $k_2$ | 2 | 3 | 4 | 5 |
|---|---|---|---|---|
| 2 | 100% | 90.6% | 69% | 40.6% |
| 3 | 91.6% | 55.2% | 15.0% | 6.2% |
| 4 | 80.0% | 19.4% | 3.2% | 1.0% |
| 5 | 48.4% | 8.2% | 2.0% | 0.0% |

Table 5: Percent correct for multiplying a $k_1$-digit by $k_2$-digit number.

## 5 Related Work

It is well understood that the form of a prompt can greatly affect the results from a LLM as a "few-shot learner" (Brown et al., 2020), thus giving rise to the newly minted discipline of *prompt engineering*. For

| $k_1 \times k_2$ | $k_2 \times k_1$ | $(\chi^2, p)$ |
|---|---|---|
| $3 \times 2$ | $2 \times 3$ | $(0.308, 0.578)$ |
| $4 \times 2$ | $2 \times 4$ | **(14.863, 1.15 e-4)** |
| $4 \times 3$ | $3 \times 4$ | $(3.398, 0.065)$ |
| $5 \times 2$ | $2 \times 5$ | **(6.158 , 0.0130)** |
| $5 \times 3$ | $3 \times 5$ | $(1.496, 0.221)$ |

Table 6: $\chi^2$ goodness-of-fit test comparing the results of a $k_1 \times k_2$ with a $k_2 \times k_1$ multiplication (i.e., the off-diagonal elements of Table 5).

example, (Yu et al., 2023) show that small differences in prompting for legal reasoning tasks has a significant impact on the accuracy of responses. Our results confirm these observations for a set of simple deterministic tasks but with high statistical significance.

On the output side, Bender et al. (Bender et al., 2021) note the dangers inherent in ascribing intent and meaning to utterances generated by LLMs. In particular, we (as humans) make many assumptions about communications with other humans that can easily lead us to fall prey to the fixed-effect fallacy when working with LLMs, potentially ascribing a more general capability to the LLM than actually exists. We show that even for simple tasks there are major sources of variance that are not easy to account for when working with LLMs.

Our experiments with deterministic algorithms are related to work that examines the capability of LLMs to perform deductive reasoning (Arkoudas, 2023). In these problems, as with most of the problems we consider, the LLM must attend to most every token in the input and not "hallucinate" new values that would lead to short-cut solutions to related but different problems than the one given. In contrast to our experiments, Arkoudas engages in a conversation with the LLM about each of the deductive problems he poses, where the LLM often proceeds to contradict itself upon getting a wrong answer. Indeed, the ad-hoc reporting of conversations with an LLM is fairly widespread (Bubeck et al., 2023) but does not rise to the level of a controlled experiment where one can make statistically significant statements. Of course, for many complex tasks it may be difficult to perform the deeper analysis we performed here for simpler tasks.

Others have observed that LLM performance degrades when the input to the LLM grows in size (within the limits of the LLM's context window), as we have shown here. Interestingly, Liu et al (Liu et al., 2023) find that information that is at the beginning or end of the context window has more influence on LLM performance, even for simple queries that ask the LLM a question whose answer is somewhere in the input. That is, the position of information is another source of variance, as we saw in the simple prompt rewording of Table 1, where the major change was to swap the position of the input list and query (wordings #1 and #2).

Wu et al demonstrate considerable performance sensitivity for a series of tasks (Wu et al., 2023). In exploring counter-factual tasks they conclude that LLMs "rely on narrow, non-transferable procedures for task-solving." Dziri et al explore failures of LLMs on seemingly trivial tasks (Dziri et al., 2023). They are especially interested in compositional tasks. They suggest that transformers often fail since they exploit linearized patch matching rather than any multi-step reasoning, and that errors propagate in a fashion that compounds. Schaeffer et al suggest that the often-discussed emergent properties of LLMs are an artifact of the metrics chosen rather than any fundamental improvement (Schaeffer et al., 2023): "For a fixed task and a fixed model family, the researcher can choose a metric to create an emergent ability or choose a metric to ablate an emergent ability."

Chain-of-Thought (CoT) is a prompting strategy that asks the LLM to output intermediate reasoning steps before giving the final answer. Research has found that it often improves LLM performance on complex tasks (Wei et al., 2022). It is worth further research to understand whether CoT-style prompts are more resilient to the variations shown in our study.

While the sensitivity of performance to prompt-phrasing has spawned the field of 'prompt engineering' efforts to quantify this sensitivity are nascent. Sclar et al examine the effect of phrasing on accuracy for multiple choice tasks using the LLaMA-2-13B model (Sclar et al., 2023). Sun et al examine zero-shot robustness on

two large standardized datasets (Sun et al., 2023). Our work extends that direction by showing sensitivity not merely to phrasing, but also input parameter, and using GPT-4 (i.e., a far larger model than used in (Sclar et al., 2023; Sun et al., 2023)). In focusing on tasks with arbitrarily large parameter spaces (e.g., counting objects in lists) we avoid many of the concerns that some variant of a task has been seen in training.

Standardized exams are often used to demonstrate LLM's capabilities. For example, studies has shown GPT-4 achieving the passing criteria of the Japanese Medical Licensing Examination (JMLE) (Takagi et al., 2023), the Uniform Bar Examination (UBE) (Katz et al., 2023), and the US Medical Licensing Examination (USMLE) (Nori et al., 2023). Knowing that even basic tasks are sensitive to trivial variations, it is legitimate to question whether the variations between a new version of an exam and its previous versions primarily focus on factors sensitive for humans, but neglect others that can be sensitive only for LLMs.

Yarkoni (Yarkoni, 2022) argues that the problem of improper generalization goes far beyond the language issue. He suggests that confusing fixed effects for random ones is the source of many of the replication failures in the social sciences.

Elazar et al explore the consistency of responses under rephrasing of various LLMs (Elazar et al., 2021). They explore general knowledge and factual questions rather than the arithmetic tasks we explore. Their findings, that all of the LLMs studied have poor consistency, are largely corroborated by our work.

Lu et al study the effect of ordering on the performance of few-shot prompts (Lu et al., 2021). They find that permuting the order in which examples in a few-shot prompt are presented can make the difference between state-of-the-art and radnom performance.

# 6 Discussion

We've shown in Section 4, the risk that measured performance with a specific prompt fails to generalize to equivalent versions of the task. This work complements others that have documented the brittleness of GPT-4's performance (see related work in Section 5). However, as far as we know, ours is the first to explore tasks with several different conditions and sufficient statistical power to rule out sampling noise as the source of observed variation. This allows us to state with some confidence that minor modifications have potentially enormous effects on measured capabilities. This problem is entirely orthogonal to the frequently mentioned difficulty with hallucinations.

Every measurement experiment comes with decisions about which factors might affect the output, and which should make no difference. Many of these decisions are implicit, and informed by our intuition and experience of the world. Since LLMs emulate many human capabilities it is tempting to use intuitions about humans to guide decisions about which factors should make no difference to LLM measurements. A key finding of this paper is that this assumption leads to errors that can be significant enough to invalidate claims. Bender observes that we've made "machines that can mindlessly generate text, but we haven't learned how to stop imagining the mind behind it." We suggest that the dangers of anthropomorphizing LLMs includes not just over-interpreting their capabilities, but also imagining that their robustness to variation resembles that of humans.

An interesting direction for future work is whether we can derive new margin-of-error bounds. Our problem is that the presence of unexplained variance means that estimating $\delta_q = 1.96 \cdot \sqrt{q \cdot (1 - q)/N}$ misses an additive component of unknown magnitude. If rewordings of a particular task can be generated automatically then estimating their variance would allow new (albeit higher) estimates of margin-of-error.

Since we warn of the risks of improper generalizations we should note the limitations of our findings. Obviously, we've explored a limited set of tasks, and a limited set of modifications of those tasks. The tasks in this paper are chosen deliberately with several criteria. First, they are deterministic tasks with easily-determined answers; this is clearly a very restricted portion of the problems to which LLMs might be applied. Second, the tasks we choose may be particularly difficult for transformer architectures. That is, the attention mechanism (Vaswani et al., 2017) decides which portions of the context window are most important in predicting the next token; however, for tasks like counting, sorting, etc., all words in the target list are important. Third, our prompts ask the questions in a concise and direct manner, without an attempt to

guide the LLM to give a Chain-of-Thought response. Finally, we emphasize that, while we have tested on GPT-4 and several other models, the problems we observe might be reduced or eliminated in later models.

# 7 Conclusion

We have demonstrated that GPT-4 performance on simple tasks shows sensitivity to trivial modifications and that this error can be enough to invalidate claims of capabilities. Despite the limited scope of our experiments, we believe our findings point to a largely-ignored source of error that potentially affects evaluation of LLM capabilities on all tasks. That is, on every task we've considered we've found that trivial modifications introduce variance that invalidates the usual margin-of-error estimates. Our evidence doesn't rule out the possibility that the problem might be larger, or smaller, or negligible on some other tasks. However, deciding that this source of error can be ignored for a given capability comes with a burden-of-proof, and is something that should be demonstrated empirically, rather than just assumed.

We find that, even when modifications are trivial and make no difference to human performance on a task, we cannot assume that the same is true of LLM performance. In the absence of evidence to the contrary, measurements of LLM task-accuracy cannot be assumed to generalize beyond the precise conditions studied.

# 8 Broader impact statment

This paper presents work whose goal is to advance the field of Machine Learning. There are many potential societal consequences of our work, none which we feel must be specifically highlighted here.

# A Appendix

Here we revisit accuracy measurements for the counting task studied in Section 4.2 but using the GPT-3.5, Mistral Instruct 7B Q4 and Llama 3 8B Q4 models. GPT-3.5 was accessed via the openai API. The Mistral and Llama models were run locally using versions with quantized coefficients. Each cell in each table represents 500 trials.

The results for these models are show in Tables 7, 8 and 9. Each of these might be compared with Table 1. As can be seen, the same pattern observed in Section 4.2 holds: the null hypothesis (that accuracy in the various conditions do not differ significantly) is robustly rejected in a majority of cases.

| rLen | Wording #1 Wts=[0.5,0.5] mango/peach | Wording #1 Wts=[0.5,0.5] airedale/ aspidistra | Wording #1 Wts=[0.7,0.3] mango/peach | Wording #2 Wts=[0.5,0.5] mango/peach | Comp. Cols(1,2) $(\chi^2, p)$ | Comp. Cols(1,3) $(\chi^2, p)$ | Comp. Cols(1,4) $(\chi^2, p)$ |
|---|---|---|---|---|---|---|---|
| 10 | 78.2% | 68.8% | 51.6% | 89.4% | **(10.86, 9.81e-04)** | **(76.49, 2.22e-18)** | **(22.28, 2.35e-06)** |
| 15 | 61.8% | 35.4% | 19.6% | 55.6% | **(68.7, 1.15e-16)** | **(182.72, 1.23e-41)** | (3.71, 5.40e-02) |
| 20 | 17.8% | 7.4% | 6.8% | 33.0% | **(23.62, 1.17e-06)** | **(27.03, 2.00e-07)** | **(29.69, 5.08e-08)** |
| 30 | 12.2% | 5.0% | 6.8% | 13.8% | **(15.58, 7.89e-05)** | **(7.86, 5.05e-03)** | (0.43, 5.10e-01) |
| 40 | 7.8% | 2.2% | 2.4% | 8.8% | **(15.35, 8.94e-05)** | **(13.97, 1.86e-04)** | (0.21, 6.47e-01) |

Table 7: GPT-3.5 Percent correct for counting the occurrences of a length-`rLen` list with two items chosen uniformly-at-random. Performance decays rapidly with list length. On the right-hand side of the table we present $\chi^2$ tests comparing the results of the first condition with each of the others. This test evaluates the null hypothesis that the answers in the various conditions are drawn from the same distribution. Boldface entries are cases where $p < 0.05$ and we reject the null hypothesis. The null hypothesis is robustly rejected for almost all lengths and conditions.

# References

Konstantine Arkoudas. GPT-4 Can't Reason. *arXiv preprint arXiv:2308.03762*, 2023.

| rLen | Wording #1 Wts=[0.5,0.5] mango/peach | Wording #1 Wts=[0.5,0.5] airedale/ aspidistra | Wording #1 Wts=[0.7,0.3] mango/peach | Wording #2 Wts=[0.5,0.5] mango/peach | Comp. Cols(1,2) $(\chi^2, p)$ | Comp. Cols(1,3) $(\chi^2, p)$ | Comp. Cols(1,4) $(\chi^2, p)$ |
|---|---|---|---|---|---|---|---|
| 10 | 37.6% | 59.2% | 28.6% | 40.8% | **(45.84, 1.28e-11)** | **(8.74, 3.11e-03)** | (0.94, 3.31e-01) |
| 15 | 14.0% | 30.8% | 15.6% | 35.6% | **(39.63, 3.07e-10)** | (0.39, 5.33e-01) | **(61.39, 4.68e-15)** |
| 20 | 16.8% | 6.2% | 22.4% | 40.2% | **(26.57, 2.54e-07)** | **(4.63, 3.15e-02)** | **(66.03, 4.43e-16)** |

Table 8: Mistral Percent correct for counting the occurrences of a length-`rLen` list with two items chosen uniformly-at-random. Performance decays rapidly with list length. On the right-hand side of the table we present $\chi^2$ tests comparing the results of the first condition with each of the others. This test evaluates the null hypothesis that the answers in the various conditions are drawn from the same distribution. Boldface entries are cases where $p < 0.05$ and we reject the null hypothesis. The null hypothesis is robustly rejected for a majority of lengths and conditions.

| rLen | Wording #1 Wts=[0.5,0.5] mango/peach | Wording #1 Wts=[0.5,0.5] airedale/ aspidistra | Wording #1 Wts=[0.7,0.3] mango/peach | Wording #2 Wts=[0.5,0.5] mango/peach | Comp. Cols(1,2) $(\chi^2, p)$ | Comp. Cols(1,3) $(\chi^2, p)$ | Comp. Cols(1,4) $(\chi^2, p)$ |
|---|---|---|---|---|---|---|---|
| 10 | 37.4% | 25.2% | 31.2% | 23.0% | **(16.74, 4.28e-05)** | **(3.99, 4.57e-02)** | **(23.91, 1.01e-06)** |
| 15 | 11.4% | 4.2% | 5.8% | 2.6% | **(17.03, 3.67e-05)** | **(9.27, 2.32e-03)** | **(28.4, 9.85e-08)** |
| 20 | 2.2% | 0.6% | 0.4% | 0.0% | (3.55, 5.96e-02) | **(4.99, 2.55e-02)** | **(9.19, 2.43e-03)** |

Table 9: Llama Percent correct for counting the occurrences of a length-`rLen` list with two items chosen uniformly-at-random. Performance decays rapidly with list length. On the right-hand side of the table we present $\chi^2$ tests comparing the results of the first condition with each of the others. This test evaluates the null hypothesis that the answers in the various conditions are drawn from the same distribution. Boldface entries are cases where $p < 0.05$ and we reject the null hypothesis. The null hypothesis is robustly rejected for almost all lengths and conditions.

Emily M Bender, Timnit Gebru, Angelina McMillan-Major, and Shmargaret Shmitchell. On the dangers of stochastic parrots: Can language models be too big? In *Proceedings of the 2021 ACM conference on fairness, accountability, and transparency*, pp. 610–623, 2021.

Tom B. Brown et al. Language models are few-shot learners. In Hugo Larochelle, Marc'Aurelio Ranzato, Raia Hadsell, Maria-Florina Balcan, and Hsuan-Tien Lin (eds.), *Advances in Neural Information Processing Systems 33*, 2020.

Sébastien Bubeck, Varun Chandrasekaran, Ronen Eldan, Johannes Gehrke, Eric Horvitz, Ece Kamar, Peter Lee, Yin Tat Lee, Yuanzhi Li, Scott Lundberg, et al. Sparks of artificial general intelligence: Early experiments with GPT-4. *arXiv preprint arXiv:2303.12712*, 2023.

Herbert H Clark. The language-as-fixed-effect fallacy: A critique of language statistics in psychological research. *Journal of verbal learning and verbal behavior*, 12(4):335–359, 1973.

Edmund B Coleman. Generalizing to a language population. *Psychological Reports*, 14(1):219–226, 1964.

Nouha Dziri et al. Faith and fate: Limits of transformers on compositionality. *arXiv preprint arXiv:2305.18654*, 2023.

Yanai Elazar, Nora Kassner, Shauli Ravfogel, Abhilasha Ravichander, Eduard H. Hovy, Hinrich Schütze, and Yoav Goldberg. Measuring and improving consistency in pretrained language models. *CoRR*, abs/2102.01017, 2021. URL https://arxiv.org/abs/2102.01017.

Andrew Gelman. Analysis of variance—why it is more important than ever. *Ann. Statist.*, 33(1):1–53, 2005.

Andrew Gelman and Jennifer Hill. *Data analysis using regression and multilevel/hierarchical models*. Cambridge University Press, 2006.

Ahmad Ghazal, Tilmann Rabl, Minqing Hu, Francois Raab, Meikel Poess, Alain Crolotte, and Hans-Arno Jacobsen. Bigbench: Towards an industry standard benchmark for big data analytics. In *Proceedings of the 2013 ACM SIGMOD international conference on Management of data*, pp. 1197–1208, 2013.

Dan Hendrycks, Collin Burns, Steven Basart, Andy Zou, Mantas Mazeika, Dawn Song, and Jacob Steinhardt. Measuring massive multitask language understanding. *arXiv preprint arXiv:2009.03300*, 2020.

Daniel Martin Katz, Michael James Bommarito, Shang Gao, and Pablo Arredondo. GPT-4 passes the bar exam. *Available at SSRN 4389233*, 2023.

Nelson F Liu, Kevin Lin, John Hewitt, Ashwin Paranjape, Michele Bevilacqua, Fabio Petroni, and Percy Liang. Lost in the middle: How language models use long contexts. *arXiv preprint arXiv:2307.03172*, 2023.

Yao Lu, Max Bartolo, Alastair Moore, Sebastian Riedel, and Pontus Stenetorp. Fantastically ordered prompts and where to find them: Overcoming few-shot prompt order sensitivity. *CoRR*, abs/2104.08786, 2021. URL https://arxiv.org/abs/2104.08786.

R. Thomas McCoy, Shunyu Yao, Dan Friedman, Matthew Hardy, and Thomas L. Griffiths. Embers of autoregression: Understanding large language models through the problem they are trained to solve. *arXiv preprint arXiv:2309.13638*, 2023.

Harsha Nori, Nicholas King, Scott Mayer McKinney, Dean Carignan, and Eric Horvitz. Capabilities of GPT-4 on medical challenge problems. *arXiv preprint arXiv:2303.13375*, 2023.

Rylan Schaeffer, Brando Miranda, and Sanmi Koyejo. Are emergent abilities of large language models a mirage?, 2023.

Melanie Sclar, Yejin Choi, Yulia Tsvetkov, and Alane Suhr. Quantifying language models' sensitivity to spurious features in prompt design or: How i learned to start worrying about prompt formatting. *arXiv preprint arXiv:2310.11324*, 2023.

Jiuding Sun, Chantal Shaib, and Byron C Wallace. Evaluating the zero-shot robustness of instruction-tuned language models. *arXiv preprint arXiv:2306.11270*, 2023.

Soshi Takagi, Takashi Watari, Ayano Erabi, Kota Sakaguchi, et al. Performance of GPT-3.5 and GPT-4 on the japanese medical licensing examination: comparison study. *JMIR Medical Education*, 9(1):e48002, 2023.

John Robert Taylor and William Thompson. *An introduction to error analysis: the study of uncertainties in physical measurements*, volume 2. Springer, 1982.

Ashish Vaswani, Noam Shazeer, Niki Parmar, Jakob Uszkoreit, Llion Jones, Aidan N Gomez, Łukasz Kaiser, and Illia Polosukhin. Attention is all you need. *Advances in neural information processing systems*, 30, 2017.

Jason Wei, Xuezhi Wang, Dale Schuurmans, Maarten Bosma, Fei Xia, Ed Chi, Quoc V Le, Denny Zhou, et al. Chain-of-thought prompting elicits reasoning in large language models. *Advances in Neural Information Processing Systems*, 35:24824–24837, 2022.

Zhaofeng Wu, Linlu Qiu, Alexis Ross, Ekin Akyürek, Boyuan Chen, Bailin Wang, Najoung Kim, Jacob Andreas, and Yoon Kim. Reasoning or reciting? exploring the capabilities and limitations of language models through counterfactual tasks, 2023.

Tal Yarkoni. The generalizability crisis. *Behavioral and Brain Sciences*, 45:e1, 2022.

Fangyi Yu, Lee Quartey, and Frank Schilder. Exploring the effectiveness of prompt engineering for legal reasoning tasks. In Anna Rogers, Jordan L. Boyd-Graber, and Naoaki Okazaki (eds.), *Findings of the Association for Computational Linguistics: ACL 2023*, pp. 13582–13596, 2023.

