# OpenReview forum: "Can We Count on LLMs? The Fixed-Effect Fallacy and Claims of GPT-4 Capabilities"
_TMLR — Accepted by TMLR_

### Review · Reviewer_YBXy · 2024-06-27

**Summary Of Contributions:**

This paper offers an evaluation of language models on synthetic tasks which admits simple generation and evaluation of tasks. Under this benchmark, the paper measures model performance with different prompting strategies (simple rewrites of the input or slightly varied instructions). Slightly changing the prompt strategy significantly changes benchmark performance, indicating that language models are brittle to slight details in the input that we don't expect to affect humans.

**Audience:**

Yes

**Claims And Evidence:**

No

**Requested Changes:**

Please see above weaknesses for more detail. Weakness 1 is the main reason for me submitting "No" to Claims and Evidence, and I am open to being corrected on this point.

**Strengths And Weaknesses:**

### Strengths

1. The paper evaluates models on synthetic tasks where a) it is easy to verify model output b) it is easy to generate many instances of the problem for statistical significance and c) it is clear which prompting details we hope would not affect the model.
2. The paper provides ample introduction to the fixed effects fallacy so the reader isn't left confused.
3. The paper adds extra evidence to the large body of work showing that language models are incredibly brittle to their choice of prompts/input style

### Weaknesses

1. [Major] I am convinced that the model should be robust to minor differences in the wording of the prompt. Some of the variations, such as replacing ints with floats and flipping multiplication, seem like major changes (and increases in difficulty) to the task. It is unclear to me that we should expect models or humans to perform equally on such tasks. Obviously a human evaluating would be gold standard, but I do not expect that from this work. What is the work's rationale on why this preserves the task?
2. When you evaluate accuracies for different prompts, do you use the same randomly generated inputs for both prompts (paired t-test) or do you use different freshly generated inputs for different prompts? Might be an important detail to clarify
3. I believe there are many other works in the literature that quantify variance due to prompting (I can recall https://arxiv.org/abs/2102.01017 and https://arxiv.org/abs/2104.08786), I would appreciate more citations to such works.
4. The point that this paper tries to establish (that models are brittle to the prompt, and this means that model "understanding" shouldn't be generalized beyond a prompt) is important but also relatively (in my opinion) well-known by the community. Furthermore, if we know that one prompt performs better, we can always use the language model with that specific prompt.
5. Section 3 feels very disjoint from Section 2. Is there actually a direct correspondence? I don't see one, but if there it would be nice to share notation somehow. I wonder if there could be a closer connection to underscore the paper's point.

---

> ### Author Response · Authors · 2024-07-15
> **Thank you for your comments**
>
> With respect to your numbered points:
>
> 1.	We don’t claim that humans would have identical performance across all conditions, however, we think it reasonable to expect similar performance. For example, when we say a child has or has not the ability to count we do not restrict that statement by details about the thing-to-be-counted, object frequency, or question wording. That is, ‘counting’ is regarded as an ability that should be independent of these particulars (even if there might be some actual variation if measured).
>
> We agree that processing 2-digit integers is probably easier than 5-6 digits floats for humans, but again we claim that our intuitions about what might be easier for humans offers and extremely poor guide. Eg, in Table 2 we might argue that the large difference between columns 1 and 3 is caused by name-value lists being harder than value lists; however, in Table 3 a large difference is observed in exactly the opposite direction (name-value lists proving easier than values alone).
>
> Thus our claim is that intuitions based on observing human performance are a very poor guide, and accuracy numbers cannot be assumed to generalize to slightly modified versions of the task.
>
> 2.	We will clarify that we generated fresh input lists for each prompt. Thank you for the suggestion.
> 3.	Thank you for the suggest related work; we’ll add these and any others we find.
> 4.	We agree that general procedures for improving accuracy would be enormously helpful; however, we don’t currently appear to have a way of doing so reliably. Chain-of-thought appears to sometimes help, but sometimes hurt. As mentioned in the answer to #1: operations over name-value lists sometimes produce worse answers (Table 2) and sometimes better (Table 3).
> 5.	We will try to make the connection clearer.

---

### Review · Reviewer_Z8io · 2024-07-01

**Summary Of Contributions:**

This paper evaluates the performance of LLMs on deterministic tasks, revealing that small changes in query phrasing or input data composition can cause large performance changes. The study highlights the fixed-effect fallacy, where observations are overgeneralized. This shows a potential problem in evaluating LLMs: sensitivity to small changes can make general claims about their abilities unreliable. Finally, the authors conclude that measuring LLM capabilities should consider these sensitivities, suggesting current evaluation methods may not capture LLM behavior nuances.

**Audience:**

Yes

**Claims And Evidence:**

Yes

**Requested Changes:**

See weaknesses

**Strengths And Weaknesses:**

**Strengths**

- This paper investigates an interesting problem. In practice, minor changes in query phrasing or input data composition can cause significant performance fluctuations in LLMs, challenging the widely accepted assumption that such variations should not affect the outcome.

- The experiment ensures reliable test results through carefully designed control conditions. The authors conducted sufficient trials (500 per condition) to detect statistically significant differences, supporting the assessment of LLM performance volatility and sensitivity.
-----

**Weaknesses**
- This paper mainly focuses on basic computational tasks such as counting, sorting, and multiplication. These tasks may not fully represent the performance of LLMs on more complex or abstract tasks.
- The study mainly examines the performance of GPT-4 and does not consider different types of LLMs or how various architectures may exhibit different characteristics and capabilities. Therefore, it is questionable whether the observations in this paper are generalizable across LLMs.
- Lack of valuable insight for model mechanisms. The paper discusses variability in LLM performance but does not explore the internal mechanisms causing these fluctuations.

---

> ### Author Response · Authors · 2024-07-15
> **Thank you for your comments**
>
> With respect to the weaknesses you identify:
> - We will clarify that we do not claim these tasks are representative of the set of tasks to which LLMs might be applied. This might fit well in the intro paragraph where we explain that tasks were selected based on the requirements of being parameterized by a population, and having answers that should be easy to parse. Thanks for the suggestion.
> - We can include an appendix with data from GPT-3.5, Mistral and Llama that shows the problem is not restricted to GPT-4.
> - We don’t have insight into the mechanism that cause the problems. We agree this is interesting and an important piece of fixing. Our hope is that drawing attention to the problem and making our data available may encourage others to also investigate.

---

### Review · Reviewer_5awU · 2024-07-05

**Summary Of Contributions:**

This paper conducts a thorough experimental study on LLM evaluation. GPT-4 is used in the experimental study on a few concrete and deterministic tasks, e.g., element counting, sorting, etc. The major observation is that trivial modifications to the prompt lead to noticeably different outputs from the LLM. The authors thus conclude that efforts to quantify LLM capabilities easily succumb to the language-as-fixed-effect fallacy, where experimental observations are improperly generalized beyond what the data supports.

**Audience:**

Yes

**Broader Impact Concerns:**

I don't find any Broader Impact Concerns in this manuscript.

**Claims And Evidence:**

Yes

**Requested Changes:**

Please see "Weaknesses" above for more detailed information. The requested changes are summarized below:

- Add more evaluation results on LLMs like Claude 3.5 and GPT-4o.
- Add more evaluation results on LLMs with different architectures like Mamba-2 and RWKV.
- Study how the generalization error evolves over time, e.g., in GPT-3, GPT-3.5, GPT-4, and GPT-4o.
- Suggest some potential approaches to mitigate the observed generalization error.

**Strengths And Weaknesses:**

Strengths:
- This paper is well-written and well-motivated in general.
- Studying the generalization of the outputs and evaluation of LLMs has potential practical impact.
- The experimental results are convincing.

Weaknesses:
- The paper claims to study the generalization of LLM evaluation; however, only GPT-4 is studied as a candidate model. I wonder if the observations generalize to other strong LLMs, e.g., Claude 3.5 and GPT-4o.
- It would also be interesting to study LLMs with different architectures, e.g., SSMs like Mamba-2 and Jamba. This would help to determine if architectural innovation helps to mitigate the generalization error observed in the paper.
- Another interesting point to study is how the generalization error evolves over time, e.g., in GPT-3, GPT-3.5, GPT-4, and GPT-4o. Assuming the major source of performance improvement in LLMs is the scale of data and model sizes, I wonder if scaling also helps with generalization.
- This paper proposes an observation on an issue but does not suggest any potential approaches for improving it. What kind of methods can we leverage to improve the generalization of LLMs?

---

> ### Author Response · Authors · 2024-07-15
> **Thank you for your comments**
>
> Reviewer 3:
> Thank you for your comments. With respect to the requested changes you identify:
> - We can include and appendix with data from GPT-3.5, Mistral and Llama that shows the problem is not restricted to GPT-4.
> - Unfortunately, we do not at present have good guidance on how this problem can be mitigated. We agree this is an important avenue of future research. We believe that even the lesser goal of determining when an LLMs answers are likely to be less reliable would be a valuable contribution.

---

### Comment · Action_Editor_Aat7 · 2024-06-07
**Assignment Acknowledgement**

Dear Reviewer Z8io,

Could you please post the assignment acknowledgement and let us know if you are able to review this submission soon?

Thank you!
AE

---

### Comment · Action_Editor_Aat7 · 2024-09-28

Dear Authors,

Can you point me to the minor revisions made in the camera-ready version as requested by the reviewers? I'm seeing the results of GPT-3.5, Mistral, and Llama in the Appendix, but not those of stronger/latest models like GPT-4o/Claude 3.5

Thanks,
AE

---

> ### Author Response · Authors · 2024-09-28
> **Reason we did not include gpt-4o, Claude 3.5**
>
> Thank you for your comment.
>
> We did not include data for gpt-4o or Claude 3.5. We point out that this paper was submitted on May 17-th; gpt-4o was released only 4 days earlier on May 13-th, and Claude 3.5 one month later on June 20-th. In our response to reviewers we offered only to include data from GPT 3.5, Mistral and Llama; since none of the reviewers objected we assumed this was acceptable. Since TMLR explicitly allows authors to reduce claims rather than carry out additional experiments we'll be happy to add a remark pointing out that it is unclear whether the observations also hold for models that became available after this work was done.

---

> > ### Comment · Action_Editor_Aat7 · 2024-10-03
> >
> > Thanks for your response. We recommended the inclusion of results using GPT-4o/Claude 3.5 because we believed this could help strengthen the claim and improve the impact of the paper. Without these results, the paper is still acceptable, but the authors should at least explicitly note (somewhere in the main paper) that the claims made may not be applicable to the latest SOTA LLMs.

---

> > > ### Comment · Action_Editor_Aat7 · 2024-10-12
> > >
> > > Dear Authors,
> > >
> > > Do you plan to incorporate the note to clarify the applicability of the claims to the latest SOTA LLMs? I'm looking forward to it.

---

### Decision · Action_Editor_Aat7 · 2024-08-19

**Recommendation:** Accept with minor revision

**Comment:**

The paper studies an interesting aspect of LLMs regarding their robustness/sensitivity in small changes of the task prompt. The authors present several sets of experiments to validate their argument that LLM evaluations should take these sensitivity factors into consideration. The reviewers generally find the experiment setups to be solid and the findings to be useful. Overall, the reviewers recommend acceptance for this work.

However, since the findings reported in this work are based on GPT-4, it remains unclear how they may generalize to newer (and stronger) models like GPT-4o, Claude 3.5 and Gemini. The authors should at least incorporate a small-scale study showing the results under multiple latest models in the revision.

**Audience:**

Yes

**Claims And Evidence:**

Yes